# A Hybrid Model for Achieving Universal Safe Drinking Water in the Medium-Sized City of Bangangté (Cameroon)

Esther Laurentine Nya [1], Roger Feumba [2], Pierre René Fotsing Kwetché [3], Willis Gwenzi [4] and Chicgoua Noubactep [5,6,7,8,*]

1  Faculty of Art, Letter and Social Sciences, University of Maroua, Maroua P.O. Box 644, Cameroon; nya.esther@yahoo.fr
2  Department of Earth Sciences, Faculty of Sciences, University of Yaoundé I, Yaoundé P.O. Box 812, Cameroon; rfeumba2002@yahoo.fr
3  Laboratory of Microbiology, Université des Montagnes, Bangangté P.O. Box 208, Cameroon; prfotsingk@gmail.com
4  Biosystems and Environmental Engineering Research Group, Department of Agricultural and Biosystems, Engineering, University of Zimbabwe, Mount Pleasant, Harare P.O. Box MP167, Zimbabwe; wgwenzi@yahoo.co.uk
5  Centre for Modern Indian Studies (CeMIS), University of Göttingen, Waldweg 26, D-37073 Göttingen, Germany
6  Department of Water Environmental Science and Engineering (WESE), School of Material Energy Water and Environmental Science (MEWES), The Nelson Mandela African Institution of Science and Technology, Arusha P.O. Box 447, Tanzania
7  Faculty of Health Sciences, Campus of Banekane, Université des Montagnes, Bangangté P.O. Box 208, Cameroon
8  Department of Applied Geology, University of Göttingen, Goldschmidtstraße 3, D-37077 Göttingen, Germany
*  Correspondence: cnoubac@gwdg.de

**Abstract:** Providing everyone with safe drinking water is a moral imperative. Yet, sub-Saharan Africa seems unable to achieve "safe drinking water for all" by 2030. This sad situation calls for a closer examination of the water supply options for both rural and urban populations. Commonly, two main aspects are considered: (1) behavioural responses to available or potential water supply options, and (2) socio-economic acceptability. These aspects determine the feasibility and the affordability of bringing safe drinking water as a basic good and human right to everyone. There is a broad consensus that achieving the UN Sustainable Development Goal 6.1 is mostly a financial issue, especially in low-income settings. This communication challenges this view as water is available everywhere and affordable treatment options are well-known. It considers the decentralized water supply model as a reference or standard approach in low-income settings rather than as an alternative. Here, the medium-sized city of Bangangté in the western region of Cameroon is used to demonstrate that universal safe drinking water will soon be possible. In fact, during the colonial period, the residences of the elite and the main institutions, including the administrative quarter, churches, and hospital, have been supplied with clean water from various local sources. All that is needed is to consider everyone as important or accept safe drinking water as human right. First, we present a historical background on water supply in the colonial period up to 1980. Second, the drinking water supply systems and water demand driven by population growth are discussed. Finally, a hybrid model for the achieving of universal access to clean drinking water, and preconditions for its successful implementation, are presented. Overall, this communication calls for a shift from safe drinking water supply approaches dominated by centralized systems, and presents a transferable hybrid model to achieve universal clean drinking water.

**Keywords:** decentralized water supply; hygiene and sanitation; Sub-Saharan Africa; waterborne disease; zero-valent iron

## 1. Introduction

A global consensus exists that costly water infrastructure development is essential to attaining universal access to safe drinking water [1,2]. Accordingly, the main problem in low-income regions such as sub-Saharan Africa (SSA) is how to finance the needed water infrastructure [1,3]. In cases where some centralized infrastructure is available, it is often dilapidated and poorly maintained, leading to systemic and repeated malfunctioning [1,2,4,5]. This is due to a severe lack of investment in water infrastructure and its maintenance. Regarding decentralized infrastructure for water supply, the appropriate choice of available technologies has been reported to be the main problem [6–10]. However, the current paradigm has largely ignored the contribution of local populations in developing, implementing, and operating their own drinking water treatment systems [10–13]. In the meantime, the science of self-reliance for decentralized water supply has been established, and several affordable technologies which are truly affordable now exist [10,12]. In turn, this calls for a revisit of the dominant global approaches to clean water supply, and a need for a new model that addresses the high financial burden associated with current approaches [10,14–16].

The world is in its third wave of global development goals, with poverty reduction as the driving motivation [1]. The current UN Sustainable Development Goals (SDGs, 2015–2030) were preceded by the Millennium Development Goals (MDGs, 2000–2015) and the International Decade of Drinking Water and Sanitation (IDDWS, 1980–1990). In fact, the struggle for universal access to safe drinking water started much earlier in the 1950s [10,17,18], a period which broadly corresponds to the birth of most African nations. It is sad that Sub-Saharan Africa has never achieved any of the established goals and is not likely to achieve the SDGs, particularly SDG 6.1, which calls for "safe drinking water for all" [1,3]. In other words, despite seven decades of concerted efforts, a large proportion of the African population still lacks safe drinking water [1,16,19,20]. This lack of progress suggests that there might be some circular reasoning in the approach to solve the named problems, pointing to the need to think outside of the box and develop and pilot test new models. The historiography of colonial water supply and sanitation can provide some insights on how to think forward and overcome clean water supply problems [20–22].

During the colonial period, there were great efforts to keep the European population in the colonies healthy [22–24]. To this end, sanitary reports were periodically written, and there were guidelines for "Health in the Tropics", including the construction and operation of hospitals, military camps, and schools to maintain hygiene and safeguard human health [22]. The architecture of colonial agglomerations in Africa depicted extensive cantonments, military quarters, and housing estates that had higher levels of water and sanitation services [22,25]. The city of Bangangté in Cameroon was such a colonial agglomeration, with an administrative quarter (including a military camp and a residential quarter), two missionary stations, and one hospital, which were all supplied with safe drinking water in a decentralized manner. The rest of the population, including lower ranking civil servants and nurses, lived in the immediate neighborhoods or in remote locations, had no modern sanitation, and used water from selected natural sources. Several similar colonial agglomerations existed elsewhere in several parts of Africa during the colonial period [2,20,25]. It is well-known in other parts of sub-Saharan Africa that traditional water management systems were available and operated in parallel to the colonial system [2,20,21]. Even though the quality of drinking water was not analytically determined, the management of water sources was locally well-organized [2]. Anecdotal evidence shows that in Zimbabwe, for example, safe and reliable drinking water sources such as perennial springs and wells were revered by the whole community and strongly protected through traditional systems. The real tragedy of Africa seems to be that both systems collapsed in parallel during the 1960s. Three reasons for this collapse were: (1) the African rulers had less power to supervise the traditional systems, (2) the new African elite was not interested in managing traditional systems meant for low-income communities, and (3) the same elite group was mostly unable to maintain and further develop the colonial water

supply system while the city's population was rapidly increasing (Section 3). The present research seeks to demonstrate the validity of this hypothesis for the city of Bangangté. Moreover, it demonstrates how a 'hybrid model' based on a combination of centralized water supply in the old city and a decentralized supply in peripheral areas can be used to achieve universal safe drinking water in Bangangté within one decade. Because colonial agglomerations in several regions share several similarities, the proposed hybrid model can be extended to several other low-income settings.

The present paper is structured as follows: (1) the presentation starts with a geographical sketch of the city of Bangangté (Section 2), (2) a historical overview of the water supply up to 1980 is then given (Section 3), (3) Section 4 gives an overview of the water supply in the city from 1980 onwards, and the current challenges for safe drinking water supply, and (4) Section 5 discusses options to achieve universal safe drinking water by 2030.

## 2. An Overview of Bangangté

Located between 5°0′0″ N and 5°20′0″ N; 10°20′00″ E and 10°50′00″ E, Bangangté is the capital city of the Ndé Division in the west region of Cameroon. The city is limited to the north by Bangang-Fokam, to the south by Bangoulap, to the west by Bamena, and to the east by the Noun river. The total area of urban Bangangté is about 7 km$^2$ [26]. The city of has 10 administrative quarters (Figure 1): Famgo (Quarter 5), Famgo-Neta 1, Koptcha (Quarter 4), Mba (Quarter 3), Mfeutom (Quarter 7), Ngakoun (Quarter 6), Noumfam (Quarter 1), Nyamjeu (Quarter 2), Sagnam (Quarter 8), and Taleum [27]. Neighbourhoods such as Banékane (which hosts the Université des Montagnes), Banékouane 1, Banékouane 2, Batéla, Mandja, Nenga, Nessah, and Peudom are considered to belong to the peri-urban zone [28].

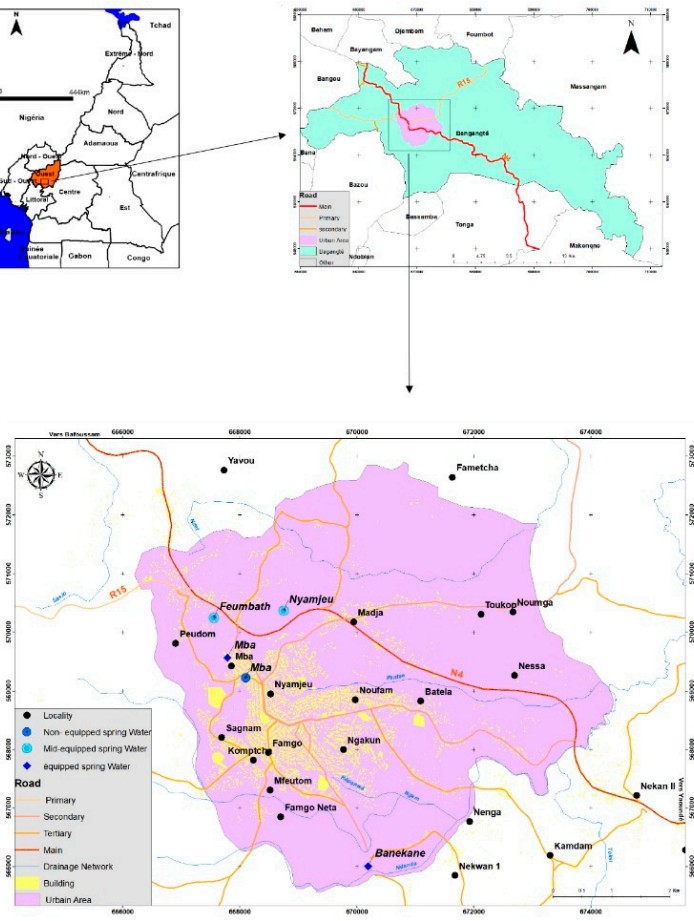

**Figure 1.** Location of the city of Bangangté in Cameroon.

The relief of Bangangté is very uneven (1340 m), which undoubtedly influences the availability and accessibility of water. The surface hydrology or drainage network is characterized by numerous rivers. However, the rivers have an irregular regime and some dry up completely in the dry season. The main river is the Ngam, which rises in the Bangoulap syenetic batholith and is colonised in its valley by Alcornéa Cordifolia and raffia. The water from this river is collected and treated by CAMWATER for the supply of the urban area. Due to the progressive lowering of the water table, which is partly linked to the *Eucalyptus* vegetation, and the decrease in rainfall, this river has experienced reduced flow or discharge [29]. Bangangté experiences a sub-equatorial Guinean climate, characterized by two dry and two wet seasons [28]. Bangangté is located in a large rainfall-deficient area [29,30]. Rainfall is low compared to other highland regions of western Cameroon. For example, Bamenda receives 3000 mm of rainfall annually, compared to 1919 mm in Dschang and only 1457 mm in Bangangté, which is located further east [29].

## 3. Drinking Water Supply System in Bangangté from the Colonial Period to 1980

This section analyses historical water infrastructure development in Bangangté. No scientific article on this issue was found in the literature. The municipality of Bangangté has no related archives. There is also no related information in the national archives in Yaoundé. Lacking published literature, we consulted a number of grey literature sources [25,31,32], master's theses [33,34], and PhD theses [5,35,36]. In cases where data were not available in grey literature and theses, we relied on personal communications and informal interviews with natives and residents of the city as informants. The informants were born before 1968, and were 13 years or older in 1980. These informants were used to provide information on the ancient water supply for the colonial period up to 1980. It is assumed that they can remember with clarity how the ancient water supply worked. The data from the informants was compared and validated with those from several other sources (Supplementary Material, in French), and the key findings are summarized here.

The colonial city of Bangangté was organized around four hills with the following infrastructure: (1) an administrative quarter comprising a city hall, a hospital, a military camp, a police station, and a primary school, all situated on the main hill of the city, and three other primary schools, each located at a missionary station (Mfeutom, Nyamjeu and Sagnam), and (2) three churches, two protestant (Mfeutom and Sagnam), and one catholic (Nyamjeu), each on a different hill. Around 1978, seven secondary schools (colleges) were available in Bangangté: one for the catholic church (Nyamjeu), two for the protestant church (Mfeutom and Sagnam), two for the government (Ngakoun and Sagnam), and two private colleges (both in Ngakoun). No information on the dates of their establishment was sought. The exact information is not important for this communication because only the three schools at the church stations of Mfeutom, Nyamjeu and Sagnam had a decentralized water supply system.

Similar to other colonial settlements in Africa and possibly elsewhere, the availability of an abundant and reliable water source was probably a prerequisite for the creation of the 'Station of Ngante' by the Germans around 1900 [25]. In fact, they "discovered" and protected the source of Feubath that is still used for decentralized water supply today (Section 3). Water from Feubath was initially used to supply the hospital and parts of the administrative quarter. Later on, the evangelic mission used the same water source for the supply of the church station of Sagnam and the related primary school and college. On the other hill, the evangelic mission used a water source situated in a valley for the supply of the station of Mfeutom. Water from the source for Mfeutom was complemented with harvested rainwater (stored in a lake) and pumped with a fuel-powered motor to a water tower at the station. On the third hill, the Catholic Mission used a local natural spring for the supply of his station comprising of a primary school, a medical center (*Ad Lucem*), and college with dormitories. Figure 2 shows a managed pipe from this source. The water pipe was installed prior to the 1980s and has never been improved. It is still largely used by

the population of Nyamjeu, of which the large majority has never been connected to the central system.

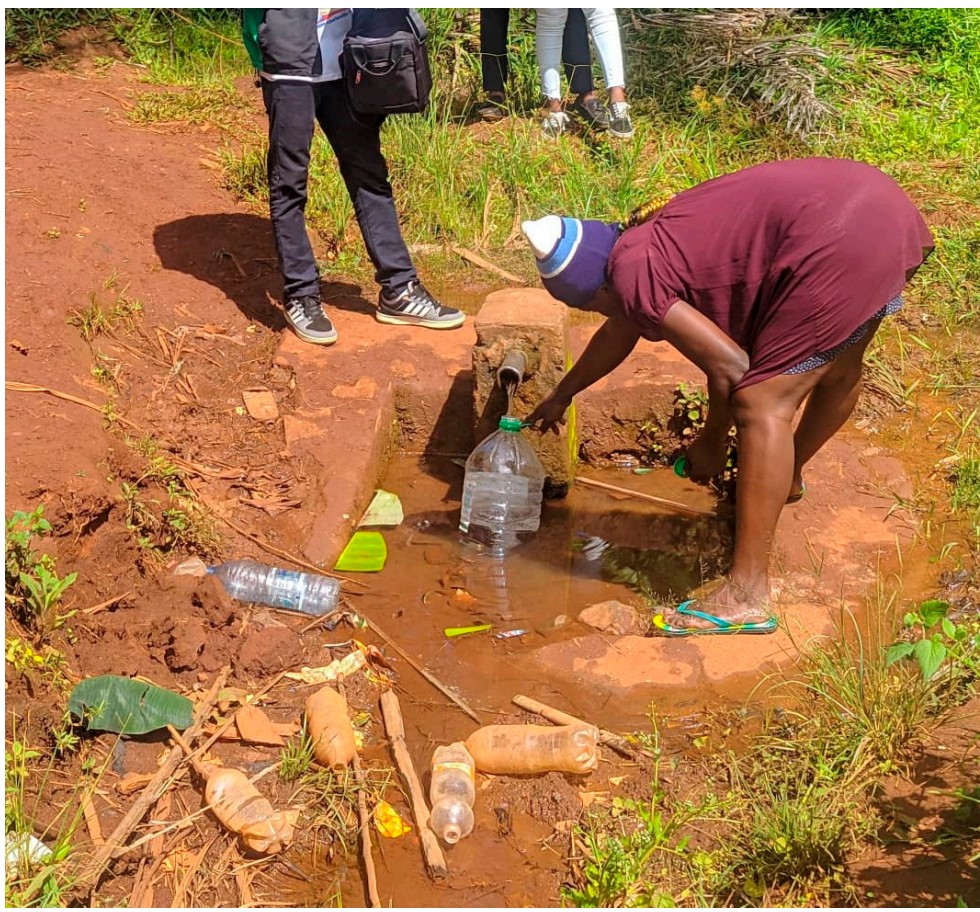

**Figure 2.** A lady fetching water from the historical spring of Nyamjeu. A crowded protected source in Mba on 6 October 2021. Photograph taken by Junior Fangang-Fanseu, October 2021.

There are contradicting reports on how the military camp was supplied with water. One source claimed that the water was drawn from the station of Mfeutom and conveyed to the camp, but the upslope location of the station relative to the water source is not consistent with this assertion. Bringing water from Mfeutom to the next hill would have required energy. For this reason, the view that the water supply of the military camp also came from the spring of Feubath (or somewhere else) is favored. Available reports indicate that water was mainly stored in underground tanks and was mechanically pumped by the end users. The military camp was closed around 1976, mainly because of the lack of a water supply. One source stated that the maintenance of the water supply system for the military camp was performed by Europeans from the missionary station of Mfeutom. These Europeans left Bangangté in the early 1970s. The rest of the urban population was supplied with water from a total of five standpipes randomly distributed in the centre of the old city. Peri-urban populations had access to non-protected natural water sources, mainly small springs, shallow wells, and ephemeral or permanent streams. Regardless of their place of living, nurses and civil servants were allowed to get water from the hospital.

This work is not assessing the colonial system, but is revealing that the colonial administration was interested in promoting and safeguarding the health of its colonial population and key service providers drawn from the African population [2,20,25]. One would have expected that after independence, the new African leaders would just consider the whole population as one, and expand the pre-existing water and sanitation systems to accommodate and integrate the formerly excluded communities to build a better continent.

Such an approach, in collaboration with local communities, could be healthy and sustainable, and ensure a safe drinking water supply [19,36,37]. The present communication is not trying to blame past and current leaders; the sole focus is to pave ways out of the current "Valley of Tears". Readers interested in details on popular causes for the lack of water in rural and urban Africa are encouraged to read two recent articles by Hope and colleagues [1,3]. For the specific case of Cameroon, a paper by Oumar and Tewari [21] gives an excellent overview of institutional and legal weaknesses in current drinking water supply approaches. These aspects are beyond the scope of the present communication. Here, a way forward is proposed despite the highlighted barriers.

In short, before 1980, the city of Bangangté was mainly supplied with three different decentralized systems. The three systems were independent and almost corresponded to a planned hill-to-hill development. Given the discharge capacity of the spring of Feubath in particular, and the evidence that even today clean water which is not collected and used just runs off, causing soil erosion and accumulating contaminants, there is a potential for a more rational use of available water resources. This is particularly true given that, even in the present era, modern water infrastructure is still lacking, while the population continues to increasing rapidly (Section 4) [2,5].

The historical perspective presented here on water infrastructure development in Bangangté serves two purposes. First, it provides practical research ideas for future studies to investigate water resources and infrastructure development in this city, and possibly elsewhere in Africa. Second, it demonstrates that a decentralized water supply system has been satisfactorily installed in Bangangté before 1980. Thus, even though the supply system was just for the elite, the potential still exists to rehabilitate and extend such an approach to cover the whole urban population.

## 4. Drinking Water Supply System in Bangangté from 1980 to 2020

Together with Mbouda and Foumbot in the western region of Cameroon, Bangangté benefited from a centralized drinking water supply network at the end of the 1970s. The corresponding catchment and treatment station is located in Batéla (Figures 1 and 3), and started its operation in 1978. Therefore, it is considered that a centralized drinking water supply was effective in Bangangté in 1980. The initial network was 50,073 km long and covered the then six structured urban districts: Famgo, Koptcha, Mba (central part), Ngakoun, Noufam, and Nyamjeu. During the past 30 years, there was a limited extension of the water supply network to some peri-urban districts like Famgo, Madja and Neuta 2. The length of the network increased from 50,073 km in 1978 to only 71,773 km in 2017, representing a 21,700 km extension (only 43%) in 40 years. By comparison, the population has increased by ninefold (Figure 4) and the area of the city has extended by 45%. This apparent mismatch between water infrastructure development and rapid population and urban growth evident in Bangangté is replicated in several other African countries [2,38]. Two interrelated factors were responsible for this huge increase of population from about 30,000 in 2007 to 90,000 in 2017: (1) the creation of higher education institutions, and (2) the accompanying migration of people seeking new opportunities and/or improved living standards [5]. A large number of these new citizens are natives from the region whose families left for the city (e.g., Bafoussam, Douala, Yaoundé) some years/decades ago. The city of Bangangté has recently witnessed the creation of four higher education institutions: (1) the University of the Mountains (Université des Montagnes-UdM 2000), (2) the Higher Institute of Technology and Commercial Studies (2008), (3) the School of Industrial Engineering (2009), and (4) the School of Health (2014). Thus, besides the students, the academic and non-academic staffs of these institutions are responsible for the observed increase in population [5,26].

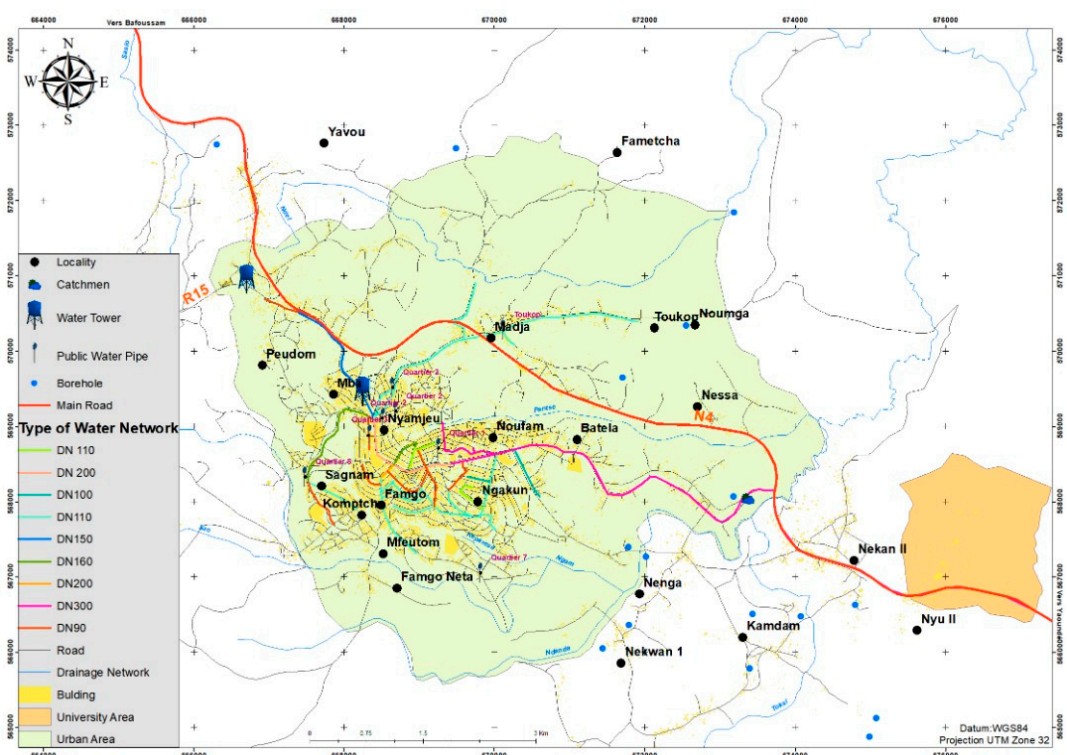

**Figure 3.** Hydraulic map of the city of Bangangté.

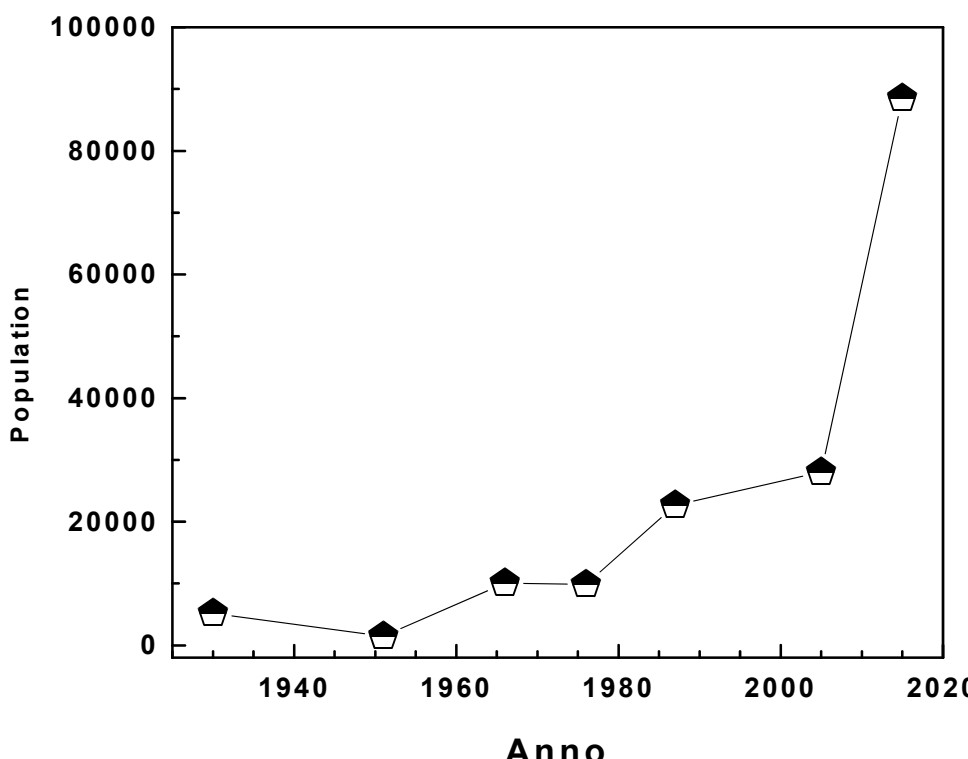

**Figure 4.** Time-dependent changes of the population in the city of Bangangté. Data compiled from refs. [27,28,31,39,40].

These 30-year-old installations can no longer produce enough water for the current population. Even with post-1980 installations, there has never been any real attempt to provide universal safe drinking water to the population of Bangangté through the

centralized system [5,37]. Table 1 gives a perfect illustration of this sad situation. It is evident that despite the increasing population (Figure 4), the treated water production capacity has actually decreased from 419,647 m$^3$ of water in 2008 to only 210,747 m$^3$ in 2013, corresponding to an approximately 50% decrease in just five years. The situation was improved later on, but the total production capacity in September 2018 was 415,748 m$^3$, still below the level of 2008. Accordingly, it is certain that, in 2021, CAMWATER produces far less water than the actual needs of the population. Although this situation is known to the municipality of Bangangté, the city authorities have only recently installed some 20 water kiosks throughout the city [5,38]. The problem of such kiosks is that they sell water from the same centralized source (i.e., CAMWATER). Thus, the water kiosks do not constitute any additional source of water, but simply redistribute the same water, albeit via different access points. This implies that the whole population of the city will suffer in the same manner whenever there is water shortage. However, the idea of partly feeding kiosks with water from different sources is already in progress. This paper advocates for using the sources that are already available and have been only marginally used during the past 30 years (Figure 2), and for extending the decentralized supply chain.

**Table 1.** Water production and distribution of the CAMWATER network in Bangangté. Data from ref. [5].

| Years | Production (m$^3$) | Distribution (m$^3$) | Loss (m$^3$) | Production Deficit Due to Power Cut (m$^3$) | Number of Active Customers | Number of Customers Served 24/24 H |
|---|---|---|---|---|---|---|
| 2005 | 292,408 | 185,257 | 107,151 | | | |
| 2006 | 385,875 | 212,032 | 173,843 | | | |
| 2007 | 442,381 | 204,914 | 237,467 | | | |
| 2008 | 419,647 | 202,130 | 217,517 | | | |
| 2009 | 323,621 | 222,381 | 101,240 | | 2658 | |
| 2010 | 290,202 | 198,986 | 91,216 | 81,338 | 2418 | |
| 2011 | 293,212 | 198,275 | 94,937 | 43,501 | 2569 | 414 |
| 2012 | 246,996 | 177,783 | 69,213 | 52,205 | 2672 | 519 |
| 2013 | 210,747 | 158,955 | 51,792 | 62,718 | 2693 | 539 |
| 2014 | 338,442 | 195,171 | 143,271 | 82,532 | 2600 | 578 |
| 2015 | 261,487 | 191,704 | 69,783 | 111,344 | 2688 | 513 |
| 2016 | 313,893 | 219,837 | 94,056 | 62,863 | 2939 | 792 |
| 2017 | 523,878 | 294,404 | 229,474 | 114,401 | 3004 | 1856 |
| 2018 | 415,748 | 230,206 | 185,542 | 56,055 | 3066 | 1928 |

Some urban districts of Bangangté are still not connected to the water supply network. This is the case of Mba (unstructured part), Nyamjeu, and Sagnam. Moreover, in the networked districts, only a small fraction (less than 30%) of the population is connected (Table 2). Additionally, even for those connected, water does not permanently flows from the taps in connected houses and kiosks [5]. Interruptions in water are very frequent and sometimes last for several days. Some interruptions are related to shortages in power production, while others are caused by water demand exceeding supply, and the breakdown of the water infrastructure [5].

Other reasons for low water production are huge losses during transport and distribution (Table 1). For example, in 2018, out of the 415,748 m$^3$ of treated water produced, 185,542 m$^3$ was lost. This accounts for an almost 45% loss, which is a huge loss considering that this is treated water. Based on the estimated average price of 0.52 US\$/m$^3$ of treated water in Bangangté, the water lost has an estimated value of USD \$96,482 Estimates show that the per capita basic water requirement for domestic use in low-income countries is about 20 L/day (i.e., 0.02 m$^3$/day) [41]. Thus, the 185,542 m$^3$ of water lost can supply an estimated 9,277,100 (9.3 million people). This points to the need for regular repair of damaged water infrastructure to reduce such water losses. Notably, the total aggregate monetary value of the water lost, and number of people who could be supplied by that water lost, is much higher if one considers the cumulative volume of water lost for various years.

**Table 2.** Main mode of drinking water supply in the city of Bangangté based on ref. [5].

| | | | Urban Area of Bangangté | | | | Total |
|---|---|---|---|---|---|---|---|
| | | | Nyamjeu | Mba | Mfeutom | Batéla | |
| Main mode of drinking water Supply | Water Network | Number | 10 | 1 | 7 | 11 | 29 |
| | | Ratio (%) | 34.5 | 3.4 | 24.1 | 37.9 | 100.0 |
| | Spring | Number | 53 | 80 | 16 | 0 | 149 |
| | | Ratio (%) | 35.6 | 53.7 | 10.7 | 0.0 | 100.0 |
| | Well | Number | 5 | 0 | 2 | 0 | 7 |
| | | Ratio (%) | 71.4 | 0.0 | 28.6 | 0.0 | 100.0 |
| | Borehole | Number | 33 | 4 | 5 | 0 | 42 |
| | | Ratio (%) | 78.6 | 9.5 | 11.9 | 0.0 | 100.0 |
| | Others | Number | 1 | 0 | 0 | 3 | 4 |
| | | Ratio (%) | 25.0 | 0.0 | 0.0 | 75.0 | 100.0 |
| Total | | Number | 102 | 85 | 30 | 14 | 231 |
| | | Ratio (%) | 44.2 | 36.8 | 13.0 | 6.1 | 100.0 |

The huge discrepancy between water needs and water produced by CAMWATER implies that the large majority of the population of Bangangté relies on informal water sources. This study advocates for a hybrid model combining the following: (1) upgrading the centralized system to better supply communities currently connected, and adding an extension to cover surrounding communities, (2) installing decentralized systems for those who cannot be deserved by centralized system within the core of the city due to lack of infrastruture or because it is too costly to do so, and (3) installing truly decentralized systems for people living in the peri-urban quarters.

A summary of the way people in Bangangté rely on informal water sources is given in Table 2, based on data of a survey of 149 households [5]. It is not surprising that 64.5% of the surveyed population obtain drinking water from springs, while only 12.5% have access to the CAMWATER network. Related informal water sources include (Table 2): boreholes, springs, streams and wells, of which some springs were exploited by colonial masters and missionaries (Section 2). Note that in the event of a water shortage, even the whole city must rely on these informal sources. In such periods, the springs of Banenkouane and Mba (Figure 5) are particularly crowded during the daytime. Figure 5 illustrates such a crowded protected source in Mba. The scientific literature contains some data on the instrumental water quality of some of the named informal water sources [26,42]. However, the quality is not discussed, as the premise is that where there is pollution, appropriate treatment shall be performed in a decentralized manner. The key is that safe drinking water provision starts and ends with instrumental water analysis.

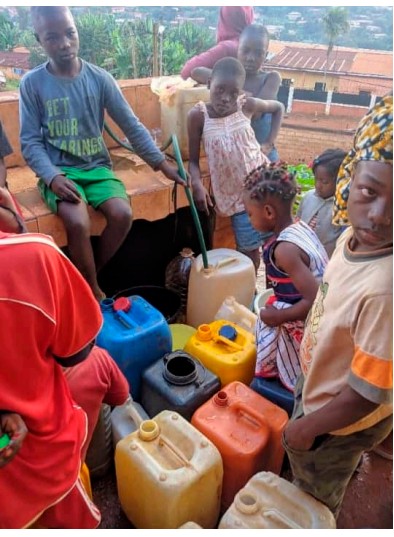

**Figure 5.** A crowded protected source in Mba on 6 October 2021. Photograph taken by Junior Fangang-Fanseu, October 2021.

## 5. Achieving the SDG 6.1 in Bangangté

This paper calls for the rediscovering of local water sources [42] and for them to be used for creating a safe drinking water supply. For this reason, two main questions to be answered are: (1) is there enough (non distant) water available to cover the current needs of the population? and (2) can the available water, if polluted, be treated in an affordable manner to drinking standards? Although groundwater (shallow wells and tube wells) is not discussed here, it is considered as a factor to be analyzed and properly treated, where necessary. There are pressing calls to broaden water testing and community-based outreach on instrumental analytical results [10,43–46].

The answer to the first question is clearly yes, given the geographic location of Bangangté (Figure 3) [42]. One potential problem is that water that flows abundantly during the rainy season could be missed during the dry season. Colleagues have recently presented a concept (Kilimanjaro Concept) to systematically harvest, partially infiltrate and abundantly store rainwater for later use [20,47–50]. The Kilimanjaro Concept has the advantage of enabling groundwater recharge as well [20,50]. An integral component of the Kilimanjaro Concept is to also harvest water from ephemeral and permanent streams. Using the colonial hill-to-hill model in Bangangté, the Kilimanjaro Concept can open a new era for a city-wide (safe drinking) water supply. Note that, in the time before 1980 (Section 3), the mission of Mfeutom was partly supplied with rainwater harvested and stored in a lake, then referred to as Etang Ministre Keutcha. Another important component of the Kilimanjaro Concept is that clean water from the springs of Banekouane I, Feubath, Nenga I and others can be put in bottles and made available in schools or sold on the market (e.g., water kiosks). This operation will reduce the runoff of clean water, which is associated with erosion and flooding. In the first instance, water for bottling can be harvested when demand and queues are low.

A detailed review of low-cost water treatment technologies is covered elsewhere [10,51], and is beyond the scope of the present study. In summary, from the recent scientific literature, the answer to the second question is also yes. In fact, there is an array of affordable do-it-yourself systems for decentralized safe drinking water treatment [10,13,15]. Among these systems, the one using metallic iron ($Fe^0$) in packed bed filters ($Fe^0$ filters) is perhaps the most accessible [52–55]. In fact, using $Fe^0$ filters, the city of Antwerp (Belgium) could supply its 200,000 inhabitants with safe drinking water for 18 months (1881/1882) without any maintenance or disruption [56,57]. That is a population more than twice the current population of Bangangté, meaning that smaller treatment units can be designed to satisfy the needs of smaller communities [58]. On the other hand, Banerji and Chaudhari [59] have developed and successfully tested a $Fe^0$ filter for the flexible water supply of small communities. This device can also be adapted to the needs of individual small communities in the peri-urban area of Bangangté. Last but not least, Tepong-Tsindé et al. [60] have recently presented, for the first time, a scalable $Fe^0$ filter using steel wool as a reactive material, and it has been able to operate for one year without any maintenance. The filter was pilot-tested in Douala, meaning that the materials and expertise can be considered to be locally available (in Cameroon). According to Huang et al. [10], $Fe^0$ filters are up to 12 times less expensive than comparable commercialized devices. Thus, they can be considered the most available and affordable technology for Bangangté. Besides the low cost of $Fe^0$ filters, lectures on decentralized safe drinking water have been given at the UdM since 2018. This means that the starting material is already locally available.

In sum, if the municipality of Bangangté equips an instrumental laboratory or encourages (or facilitates) the creation of such services, the city can assess water quality and decide on which technologies to use to make available waters potable. The focus here is on the municipality because it is the smallest political entity, and a decentralization trend has started in Cameroon [61], with specific emphasis on environmental and natural resources management, including the water supply.

The development and implementation of decentralized drinking water supply systems requires a supportive regulatory, institutional and policy framework. Yet, the non-existence

of a proper water policy (and law) in Cameroon is well-known and discussed, for example by Oumar and Tewari [21]. Their [21] conclusion was the following: "The creation of a financially autonomous water structure from which a good water policy and sound water law can be drafted to account for the present poor water management system is timely for the country". The present authors regard this study as their response to this call, directly addressed to the city of Bangangté, but adaptable to all other communities worldwide [62–66]. In particular, a roadmap is made available to associations, donors, non-governmental organizations and religious entities willing to supply (remote) small communities with safe drinking water from boreholes, lakes, rivers, springs, streams, or wells.

Coming back to the city of Bangangté, Section 4 has demonstrated that the municipality cannot supply even the old city with sufficient safe drinking water from the existing centralized system that has been available since the end of the 1970s. However, a hybrid model combining centralized and decentralized systems can cover the needs of an increasing population. Thus, the establishment of an operating hybrid model is urgently needed. The system should be monitored and maintained under the control of the city council. Ideally, each public water point should be independently analyzed (e.g., monthly) and the results made accessible to the users [43–46]. Private water providers should be responsible for the quality of water they are distributing. Accordingly, the new approach also opens markets for water analysis (quality control) and private water distribution at the ground level (community level). For example, an accredited provider or entrepreneur can specialize in supplying kindergartens and primary schools with safe drinking water.

## 6. Concluding Remarks

There have been disparities in access to safe drinking water in Bangangté since the colonial period. This research has retraced the history of the water supply in Bangangté and showed that, using the same available natural water sources, the pre-independence decentralized design can be revived and extended to cope with the need of an increasing population in an expanding city. This is a strong call for immediate action. Yet, infrastructure and technical skills are still lacking, and the existing infrastructure has received little attention to date. Moreover, insufficient resources have been allocated to address the current water supply deficits. However, the lack of financial resources applies to the design and implementation of centralized systems. The knowledge to design decentralized sustainable safe drinking water systems exists. Furthermore, Bangangté is blessed with educational centers where special skills for water management can be rapidly introduced. Clearly, all that is needed is the political will (e.g., of the municipality) to end a long lasting injustice with regard to access to safe drinking water.

The situation of Bangangté is common to most African cities, with the urban poor being disproportionately affected. Each city has its own governance, history, hydrology, population, and water resources. Accordingly, the magnitude of the challenges differs from case to case. In all African countries, occasional sectoral reforms and increased budgetary allocations to the water sector have not really improved the living standards of the poor residents with respect to access to clean drinking water and improved sanitation [3,67–70]. The mistake has been to try to create city-wide, large-scale piped water systems. Our research advocates for decentralized initiatives as affordable, adaptable, and sustainable solutions for all cases. The current SARS-CoV pandemic situation serves as a reminder that water treatment systems should not rely on imported components.

**Supplementary Materials:** The summary of the information on the ancient water supply in Bangangte in French is available online at https://www.mdpi.com/article/10.3390/w13223177/s1.

**Author Contributions:** Conceptualization, C.N., E.L.N., R.F. and W.G.; methodology, E.L.N., P.R.F.K. and R.F.; writing—original draft preparation, E.L.N., C.N. and P.R.F.K.; writing—review and editing, E.L.N., C.N., P.R.F.K. and W.G. All authors have read and agreed to the published version of the manuscript.

**Funding:** This research received no external funding.

**Institutional Review Board Statement:** Not applicable.

**Informed Consent Statement:** Not applicable.

**Data Availability Statement:** All data are included in the paper.

**Acknowledgments:** Calvin Léonard Kemadjou (Bangangté/Cameroon) and Donkou Noubactep (Cape Town/South Africa) are thanked for the valuable information on the history of water supply in Bangangté. Junior Fangang-Fanseu is thanked for technical support. We would like to thank the peer reviewers for their valuable suggestions and comments on improving this paper. We acknowledge support by the German Research Foundation and the Open Access Publication Funds of the Göttingen University.

**Conflicts of Interest:** The authors declare no conflict of interest.

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
