# Peer review of "A Hybrid Model for Achieving Universal Safe Drinking Water in the Medium-Sized City of Bangangté (Cameroon)"

_water, doi:10.3390/w13223177_

Round 1
Reviewer 1 Report
The manuscript ”A hybrid model for achieving universal safe drinking water in the medium-sized city of Bangangté (Cameroon)” has a very important topic: finding solutions for achieving the UN Sustainable Development Goal 6.1 (i.e., ensuring access to water and sanitation for all). In particular, this manuscript is focused on the detailed presentation of the case of the medium-sized city of Bangangté, in Cameroon. For the research group headed by Dr. Noubactep this work is the continuity of several recent investigations discussing the usefulness and affordability of appropriate decentralized solutions for safe drinking water provision in low-income communities.
The issues discussed in this study are very well documented, based on numerous recent references. The two tables are useful, clear and suitably captioned. The manuscript is written in a well-organized and systematic way, with a necessary overview on historical background of Bangangté water supply, as well as on the existent drinking water supply systems and water demand driven by population growth. Finally, a hybrid model based on a combination of centralized water supply (in the old city) and decentralized supply (in peripheral areas), and the preconditions for its successful implementation, were presented and discussed. The authors demonstrated with this study that such a hybrid model can be used for achieving universal access to clean drinking water in the city of Bangangté within one decade
To sum up, this manuscript fills a much-needed gap in the field of water management systems. A very large amount of work was involved in the study, and as far as I can determine, the work is solid and the findings are very interesting. Therefore, I believe the paper is appropriate for publication in Water, after addressing the following comments:
Comment 1: Page 7, lines 249-252: ”The length of the network increased from 50,073 km in 1978 to only 71,773 km in 2017, representing a 21,700 km extension (only 43 %) in 30 years. By comparison, the population has increased by 47 % (Figure 4) and the area of the city extended by 45 %.”
First: From 1978 till 2017 is a period of 40 years (not 30)
Second: From figure 4 it results that the population in 1978 was about 10,000, and in 2017 about 90,000. Thus, it is a nine-fold increase (not 47%). By the way, it would be interesting to explain in the manuscript what was the cause of the recent huge increase of population from about 30,000 in 2007 to 90,000 in 2017.
Comment 2: Figure 5: Remove the two black stripes existent at the top and bottom of figure
Author Response
Comments 1: The manuscript ”A hybrid model for achieving universal safe drinking water in the medium-sized city of Bangangté (Cameroon)” has a very important topic: finding solutions for achieving the UN Sustainable Development Goal 6.1 (i.e., ensuring access to water and sanitation for all). In particular, this manuscript is focused on the detailed presentation of the case of the medium-sized city of Bangangté, in Cameroon. For the research group headed by Dr. Noubactep this work is the continuity of several recent investigations discussing the usefulness and affordability of appropriate decentralized solutions for safe drinking water provision in low-income communities.
Many thanks for this evaluation!
Comments 2: The issues discussed in this study are very well documented, based on numerous recent references. The two tables are useful, clear and suitably captioned. The manuscript is written in a well-organized and systematic way, with a necessary overview on historical background of Bangangté water supply, as well as on the existent drinking water supply systems and water demand driven by population growth. Finally, a hybrid model based on a combination of centralized water supply (in the old city) and decentralized supply (in peripheral areas), and the preconditions for its successful implementation, were presented and discussed. The authors demonstrated with this study that such a hybrid model can be used for achieving universal access to clean drinking water in the city of Bangangté within one decade
Many thanks for this evaluation!
Comments 3: To sum up, this manuscript fills a much-needed gap in the field of water management systems. A very large amount of work was involved in the study, and as far as I can determine, the work is solid and the findings are very interesting. Therefore, I believe the paper is appropriate for publication in Water, after addressing the following comments:
Many thanks for this evaluation!
Comments 4: Page 7, lines 249-252: ”The length of the network increased from 50,073 km in 1978 to only 71,773 km in 2017, representing a 21,700 km extension (only 43 %) in 30 years. By comparison, the population has increased by 47 % (Figure 4) and the area of the city extended by 45 %.”
First: From 1978 till 2017 is a period of 40 years (not 30)
Corrected, Thanks!
Comments 5: From figure 4 it results that the population in 1978 was about 10,000, and in 2017 about 90,000. Thus, it is a nine-fold increase (not 47%). By the way, it would be interesting to explain in the manuscript what was the cause of the recent huge increase of population from about 30,000 in 2007 to 90,000 in 2017.
Two inter-related factors were responsible for this huge increase of population from about 30,000 in 2007 to 90,000 in 2017: (i) the creation of higher education institutions, and (ii) the accompanying migration of people seeking for new opportunities and/or improved living standards. A large part of this new citizens are natives from Bangangté and the Ndé division whose families left for the city some years/decades ago. The city of Bangangté has recently witnessed the creation of four (4) high education institutions: (i) the University of the Mountains (2000), (ii) the Higher Institute of Technology and Commercial Studies (2008), (iii) the School of Industrial Engineering (2009), and the School of Health (2014). Thus, beside the student, the academic and non academic staffs of this institutions are responsible for the observed increase of population.
Comments 8: Figure 5: Remove the two black stripes existent at the top and bottom of figure.
Removed, Thanks!
Many thanks for your valuable and very encouraging remarks!
Sincerely,
Dr. Noubactep
Reviewer 2 Report
The article concerns the safe drinking water in a selected city in Cameroon. It may be interesting for readers of Water. The following requests/suggestions should be taken into account to improve the quality of the manuscript.
- In my opinion it is not clear what is the novelty of this study. The article reviews the history of the water supply in the area, assesses the current state of the water supply and formulates general recommendations without leading to an increase in knowledge on the area's water supply.
- Authors should identify the gaps in research and show the key goals obtained. Authors should include limitations of their methodology, their key goals and overall, why they are making science to advance.
- In developed countries, the decision on the suitability of water for consumption is issued by the appropriate sanitary authority. Who makes such decisions in Cameroon? Can we talk about safe water without regular monitoring of water quality, control of the water level and control technical condition? Water providers must be controlled.
- New water treatment technologies should be approached with caution, especially if they are in the testing phase (Fe0 filters). Water quality is always a priority.
- Tables are not uniform.
- Figure 5 needs to be formatted (please remove the smartphone menu).
Author Response
Comments 1: The article concerns the safe drinking water in a selected city in Cameroon. It may be interesting for readers of Water.
Responses 1: Many thanks for this evaluation. We have presented on the example of Bangangte, how an hybrid system can be used to accelerate access to safe drinking water.
The following requests/suggestions (Comments 2 to Comments 7) should be taken into account to improve the quality of the manuscript.
Comments 2: In my opinion it is not clear what is the novelty of this study. The article reviews the history of the water supply in the area, assesses the current state of the water supply and formulates general recommendations without leading to an increase in knowledge on the area's water supply.
Responses 2: Novel is a pragmatic and practical approach so solve a long lasting problem. A road map for NGOs but also for municipalities. Novel is the evidence that the concept is applicable even despite lacking financial and political frameworks but would be better with those frameworks.
Comments 3: Authors should identify the gaps in research and show the key goals obtained. Authors should include limitations of their methodology, their key goals and overall, why they are making science to advance.
Responses 3: No we are not advancing science in this paper. We are synthesizing available knowledge on water management (not only treatment) to accelerate the achievement of SDG 6.1. This was possible only thanks to our diverse backgrounds (Chemical Engineering, Environmental Science, Geography, Geology, Water Chemistry). The concept for Bangangte is founded on 16 years own expertise.
Comments 4: In developed countries, the decision on the suitability of water for consumption is issued by the appropriate sanitary authority. Who makes such decisions in Cameroon? Can we talk about safe water without regular monitoring of water quality, control of the water level and control technical condition? Water providers must be controlled.
Responses 4: We have clearly addressed the weaknesses of the lack of legal and institutional frameworks while pointing out the necessity of monitoring and publishing the quality of drinking water.
Because water quality is the most important point, we have added the following “There are pressing calls to broaden water testing and based community outreach on instrumental analytical results [10,43-46].” while citing for recent calls for regular water testing. In particular, Hubbart and Gootman (2021) consider water testing as a powerful tool to address drinking water security and booster community outreach. We totally agree with them.
Comments 5: New water treatment technologies should be approached with caution, especially if they are in the testing phase (Fe0 filters). Water quality is always a priority.
Responses 5: As clearly stated in the text, Fe0 filters have been successfully used for 140 years and Indian colleagues have recently presented efficient Fe0 filters. Our own research group hat tested steel wool filters capable as operating for 1 year without any maintenance. All these information is available in the submission, together with the statement that only water quality should decide on whether treatment is necessary and which technologies are applicable. Again, Fe0 filters are presented as a candidate, also because expertise is largely available locally.
Comments 6: Tables are not uniform
Responses 6: The reviewer is right. We trust MDPI for the final format. The given information is clear to the reader.
Comments 7: Figure 5 needs to be formatted (please remove the smartphone menu).
Responses 7: Corrected, many thanks!
We hope that these clarifications address your concerns. Many thanks for your evaluation.
Sincerely,
Dr. Noubactep
Round 2
Reviewer 2 Report
Thank you for your your clarification and improvements. All the best.